# Alternate Wetting and Drying Irrigation Reduces P Availability in Paddy Soil Irrespective of Straw Incorporation

Fanxuan Kong [1], Xintan Zhang [1], Yonghe Zhu [1,2], Haishui Yang [1,2] and Fengmin Li [1,2,*]

1   College of Agriculture, Nanjing Agricultural University, Nanjing 210095, China;
    2019201012@njau.edu.cn (F.K.); xintan-zhang@foxmail.com (X.Z.); yhz@njau.edu.cn (Y.Z.);
    yanghaishui@njau.edu.cn (H.Y.)
2   Jiangsu Collaborative Innovation Center for Modern Crop Production, Nanjing Agricultural University,
    Nanjing 210095, China
*   Correspondence: fmli@njau.edu.cn

**Abstract:** Crop production is highly impacted by soil phosphorus (P) availability which is poor and susceptibly affected by soil moisture. However, how water management and straw incorporation affect paddy soil P availability is still not well known. A 40-day incubation experiment was conducted to evaluate the effects of two water management regimes: continuous flooding irrigation (CF) and alternate wetting and drying irrigation (AWD) combined with different straw addition rates (equivalent to 0, 50%, 100%, 200%, and 300% straw incorporation rates in field) on P availability in paddy soil. Water management significantly affected soil available P, microbial biomass P, total reductant, and ferrous iron. However, straw addition showed no effect on soil P availability in the short term. Compared to CF, AWD consistently decreased the soil available P content under straw addition at different rates. The main reason was that AWD increased microbial biomass for immobilizing P and decreased ferrous iron content for increasing soil P absorption, reducing available P content. In conclusion, AWD reduces available P content in paddy soil compared to CF. Water management has a more significant regulatory effect on soil P availability than straw incorporation in the field management.

**Keywords:** available P; microbial biomass P; soil moisture content; straw addition; redox condition

## 1. Introduction

Phosphorus (P) is an essential macronutrient for crop growth. However, soil available P content is usually low due to slow diffusion of phosphate anion and high fixation in soil [1]. Soil P fraction is affected by three main mechanisms: precipitation and dissolution, adsorption and desorption, and biotransformation [2]. Unlike dryland soils, the change in paddy water environment during rice production can easily affect soil P availability [3,4]. Therefore, water management in paddy plays a vital role in water use and soil nutrient availability. Compared with traditional continuous flooding irrigation (CF), alternate wetting and drying irrigation (AWD) is an effective water-saving irrigation method [5]. Moreover, field experiments have shown that moderate AWD could increase P use efficiency and rice yield [6,7]. However, few studies have confirmed whether AWD can improve soil P supply relative to CF.

A large number of studies have revealed the effect of flooding on soil P status. Flooding can change soil redox conditions and lead to the transformation of the iron form [8,9], which is an important factor controlling soil P precipitation and dissolution as well as adsorption and desorption [10,11]. Flooding decreases soil redox potential, resulting in the $Fe^{3+}$ minerals being reduced to ferrous iron, thereby releasing P from insoluble iron phosphate [12]. Additionally, some research showed that flooding increases soil P absorption and the maximum P adsorption capacity due to the increased formation of amorphous iron oxides (extracted by ammonium oxalate buffer), with many P adsorption

sites [13–15]. However, the increasing maximum P adsorption under reduced (flooded) soils was likely an experimental artifact. Its maximum P adsorption might be overestimated because the exposure of the reduced soils to aerobic conditions and the use of high P additions might result in P precipitation [16].

A cycle of AWD includes two stages (a flooding period and drying period) but few studies focus on soil P availability of the drying period. Most field trials with different water management regimes evaluate paddy soil P availability using air-dried soil collected at the harvest stage [7,17]. Additionally, some researchers compared the changes in available P and P adsorption in flooded soils and their air-dried samples [10,18]. However, this may not reflect the response of soil P status to water management regimes during rice growth. In actual field management, due to rice growth, high soil water content is still maintained in the drying period of AWD. The dry degree of soil samples may matter in evaluating soil P availability. Zhang et al. indicated that the forms of soil elements, such as iron and P, would be primarily changed in soil air drying [19]. Therefore, the actual situation of P in paddy soil may be better reflected with fresh soil samples under different water management.

The addition of organic matter, such as manure compost and decaying straw, could increase soil available P through direct P supply and reduction in P sorption [20]. Straw incorporation is a vital field management practice for rice production in China [21–23]. Previous studies showed that soil available P content could be increased by straw incorporation [24] but depended on the amount of straw returning [25]. Soil nutrient status changes dynamically with crop growth and field management practices, such as fertilization [26]. The prophase effects of straw returning on soil nutrients are closely related to crop growth. However, few studies on the effects of straw addition on soil P availability within several weeks were reported. Rakotoson et al. found that new organic matter such as straw promoted reductive dissolution of soil $Fe^{3+}$ minerals under soil flooding, thus mobilizing associated phosphate [27]. A 66-day incubation experiment revealed that straw addition could increase soil exchangeable P [12].

Microbial biomass P is an important component and the most active part of soil organic P pool. It accounts for 2–5% of the total organic P content in arable soil [28]. The microbial biomass P can quickly participate in nutrient cycling and become an essential source of plant-available P due to the fast turnover rate. The organic P immobilized by microorganisms is mineralized soon after their death. In addition, the $Fe^{3+}$-reducing bacteria number increased in flooded soil [29], releasing phosphate anions due to the microbially mediated reductive dissolution of $Fe^{3+}$ oxides [8]. Additionally, soils treated with AWD were found to be more diverse and enriched in microbial community than CF [30]. The current research on soil P availability under water regulation practices mainly focuses on the adsorption and desorption of soil P [15,31], but the relationships between microbial biomass P and soil available P under different water management regimes are still unclear.

This study explores the effects of water management and straw addition on paddy soil P availability. It clarifies the relationships between soil available P and redox status, pH, and microbial biomass P under different water management regimes. We hypothesized that (1) alternate wetting and drying irrigation reduces the availability of P in paddy soil with or without straw addition compared with continuous flooding irrigation; (2) the addition of straw improves P availability in paddy soil under two water management regimes.

## 2. Materials and Methods

### 2.1. Soils and Wheat Straws

The paddy soil (0–20 cm) was collected in Rugao city, Jiangsu Province, China (120°73′ E, 32°43′ N). The primary cropping system is rice–wheat annual double cropping. Soil water-holding capacity was 25.9%. The soil was a typical sandy loamy soil with a pH of 7.06 and contained 17.8 g/kg organic carbon, 27.8 mg/kg available P, 10.6 mg/kg available N, and 4.54 g/kg total Fe [32]. After being sieved through a 2 mm sieve and adjusted to 100% of the water-holding capacity, the fresh soil was incubated under a dark

environment at room temperature for one week to restore microbial activity. The wheat straw was dried in an oven and then chopped at 1–2 cm.

### 2.2. Experimental Set-Up

The experimental design was a factorial combination of water management and straw addition. Ninety glass bottles (upper mouth diameter: 6.3 cm, bottom diameter: 7 cm, height: 17.5 cm) were filled with equivalent to 400 g dry soil, which mixed uniformly with the corresponding wheat straws. The experiment included five straw addition rates: 0, 0.416 g, 0.832 g, 1.664 g, and 2.496 g per bottle, equivalent to 0, 50%, 100%, 200%, and 300% straw incorporation rates in the local field (according to the average soil bulk density of 0–20 cm cultivated layer of 1.3 g/cm$^3$, and the total amount of wheat straw returned to the field of 5400 kg/ha). Water management regimes included continuous flooding irrigation (CF) and alternate wetting and drying irrigation (AWD). There were ten treatments in the experiment, and every treatment contained nine bottles. All bottles were added with deionized water to 2 cm above the soil surface using a weighing method to ensure the same quality of water supply. Subsequently, they were put in an incubator at 30 °C to simulate the natural drying process for 40 days. The incubator fan's blade ensured that the temperature was uniform and the evaporated water was discharged out of the incubator. In the incubation period, CF treatments were maintained in the water layer at 1–3 cm while adding deionized water to 2 cm above the soil surface by using the weighing method when the soil water content of AWD treatments decreased at nearly the soil water-holding capacity. We define the stage when the soil surface is covered with a water layer as the flooding period of AWD. Additionally, the stage from the no-water layer on the soil surface to the decline of soil water content to the water-holding capacity is defined as the drying period of AWD. During 40-day incubation, AWD treatments suffered three wetting–drying cycles. For AWD treatments, soils were sampled once in each wetting–drying process. Soils of three replicates were sampled on the second, fourth, and sixth days of the drying period (DDP) of three wetting–drying cycles. The CF treatments were sampled at the same time as AWD treatments.

After removing the straw from soil by using tweezers and mixing soil samples uniformly, the soil water content of the AWD treatments was measured directly by the oven-drying method. Figure 1 shows the soil water content of AWD treatments at three sampling periods. The soil water content of AWD treatments was consistent in each sampling period. It is difficult for CF treatments' soils to use the same measurement methods for chemical indexes as AWD, because the soils were covered with a water layer. Therefore, soils of CF treatments were centrifuged for 10 min (3600 r/min) to remove surfaced water to make their soil water content as consistent as possible. Then, the soil water content of soil samples after centrifugation was determined by the oven-drying method. The average soil water content of CF treatments was 0.466 g g$^{-1}$.

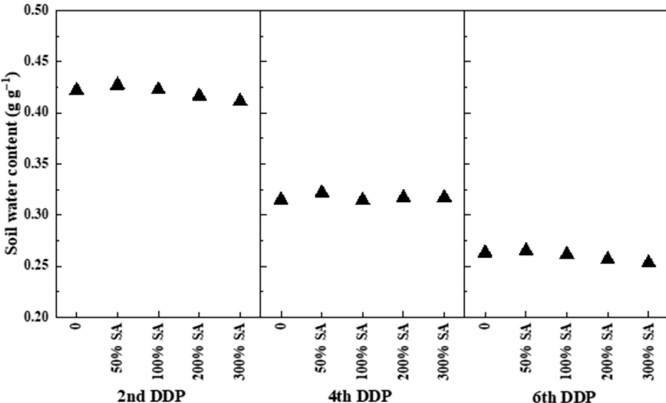

**Figure 1.** Average soil water content (three replicates) in alternate wetting and drying irrigation treatments at three sampling periods. SA, straw addition rate. DDP, day of the drying period.

### 2.3. Measurements

Each soil sample was divided into two parts. One part of fresh soil was kept in a refrigerator at 4 °C to determine available P, pH, microbial biomass P, ferrous iron, and total reductant within a week. The concentration of above chemical indexes of each fresh soil sample was calculated according to its soil water content. In contrast, another part was used to determine available P and microbial biomass P after air drying.

Soil pH was measured with a pH meter (Mettler-Toledo FE28, Zurich, Switzerland) using a soil-to-water ratio of 1:2.5 (*w/v*) ratio. Fresh soil equivalent to 10 g of dry soil was weighed, and the corresponding volume of water was taken according to the water content of each soil sample. Available phosphorus content was determined by the molybdenum antimony anti-spectrophotometric method, where the soil samples were extracted with 0.5 M $NaHCO_3$ (pH = 8.5) for 30 min and then 2.5 mL molybdenum antimony anti-color reagent was added and OD values were measured at 880 nm using a spectrophotometer [32]. The determination of microbial biomass P content followed the procedure by Brookes et al. [33]. Ferrous iron content and total reductant content were determined according to Lu [32]. Briefly, ferrous iron content was determined by the phenanthroline colorimetric method and total reductant content was measured by the potassium dichromate oxidation method after extraction with 0.1 M $Al_2(SO_4)_3$ (pH = 2.5).

### 2.4. Statistics Analysis

For each sampling period, differences in soil variables, available P of fresh soil, available P of air-dried soil, pH, microbial biomass P, ferrous iron, and total reductant, were tested using a one-way analysis of variance (ANOVA) under two water management regimes. A least significant difference (LSD, *p* = 0.05) was applied to assess the differences between the means. A two-way ANOVA was performed to assess statistical differences between straw addition and water management and their interactions for each sampling period. Normality tests were conducted for all of the above data and they were log-transformed if they did not satisfy the assumption of normality before statistical analyses. The Pearson test (two-tailed) at *p* < 0.01 was used for analyzing correlations among measured variables under two water management regimes. SPSS 20.0 (IBM, Armonk, NY, USA) was used for all those analyses. Origin Pro 21.0 (OriginLab, Northampton, MA, USA) was used to perform regression analysis and draw figures.

## 3. Results

### 3.1. Soil P Availability

Both the available P content of fresh and air-dried soil were significantly affected by water management (except fresh-soil available P content on the 2nd day of the drying period (DDP). At the same time, straw addition had no significant effects on available P content (Figure 2). Whether fresh or air-dried soil, available P content was higher in CF than in AWD. Compared with CF, AWD's average fresh soil-available P content decreased by 23.3% and 34.1% on 4th DDP and 6th DDP, respectively. Additionally, the average air-dried soil available P content of AWD decreased by 8.52%, 11.1%, and 14.3% on 2nd DDP, 4th DDP, and 6th DDP, respectively. After air drying, the available P content of soil increased. The increment was different under two water management regimes at three sampling periods, indicating that the determination using air-dried soil under different water management regimes could not reflect the P status of fresh soil.

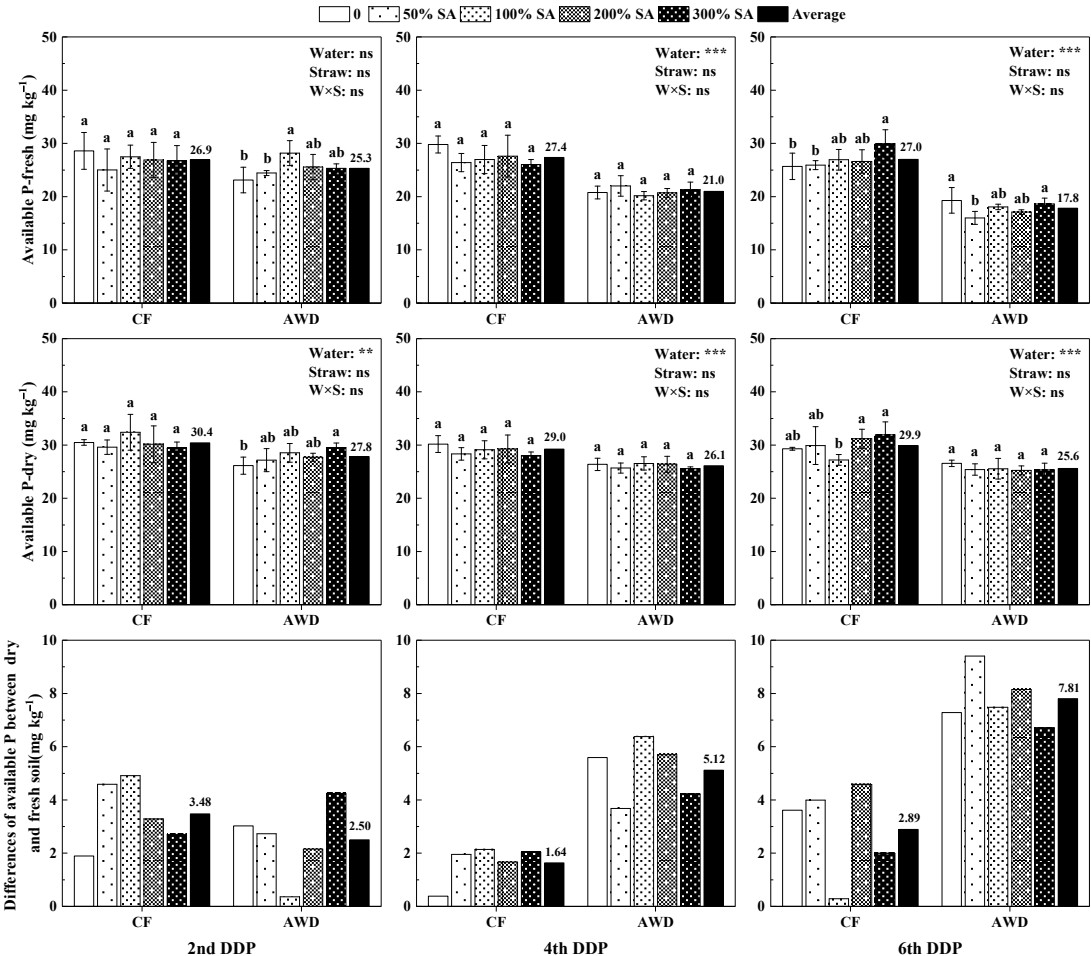

**Figure 2.** Fresh and air-dried soil available P content and differences of available P content between air-dried soil and fresh soil under different water management regimes and straw addition rates. Different letters indicate significantly different at *p* < 0.05 under the same water management regime. Data in the figures indicate means of three replicates ± standard deviation. ** means *p* < 0.01; *** means *p* < 0.001; ns means non-significant. CF, continuous flooding irrigation; AWD, alternate wetting and drying irrigation; SA, straw addition rate. DDP, day of the drying period.

### 3.2. Soil pH and Microbial Biomass P

Soil pH was significantly affected by water management and straw addition on the 2nd DDP and by their interaction on 6th DDP while only affected by water management at 40 d (Figure 3). In general, the pH of all treatments at three sampling periods was changed in 7.15–7.61, which was at intermediate pH values. Soil microbial biomass P content was significantly affected by water management at three sampling periods. It was also significantly affected by straw addition on 6th DDP and the interaction between water management and straw addition on 4th DDP (Figure 4). Relative to CF, AWD increased microbial biomass P content. AWD treatments increased microbial biomass P content by 14.1%, 49.0%, and 26.1% on average on 2nd DDP, 4th DDP, and 6th DDP, respectively. Microbial biomass P content decreased with straw addition rates on 4th DDP while increasing under CF on 6th DDP.

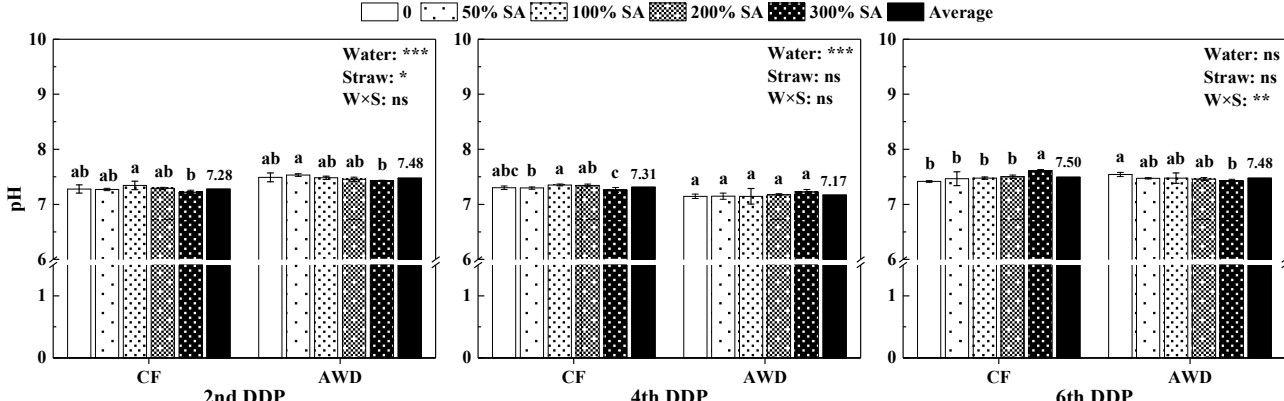

**Figure 3.** Soil pH under different water management regimes and straw addition rates. Different letters indicate significantly different at *p* < 0.05 under the same water management regime. Data in the figures indicate means of three replicates ± standard deviation. * means *p* < 0.05; ** means *p* < 0.01; *** means *p* < 0.001; ns means nonsignificant. CF, continuous flooding irrigation; AWD, alternate wetting and drying irrigation; SA, straw addition rate. DDP, day of the drying period.

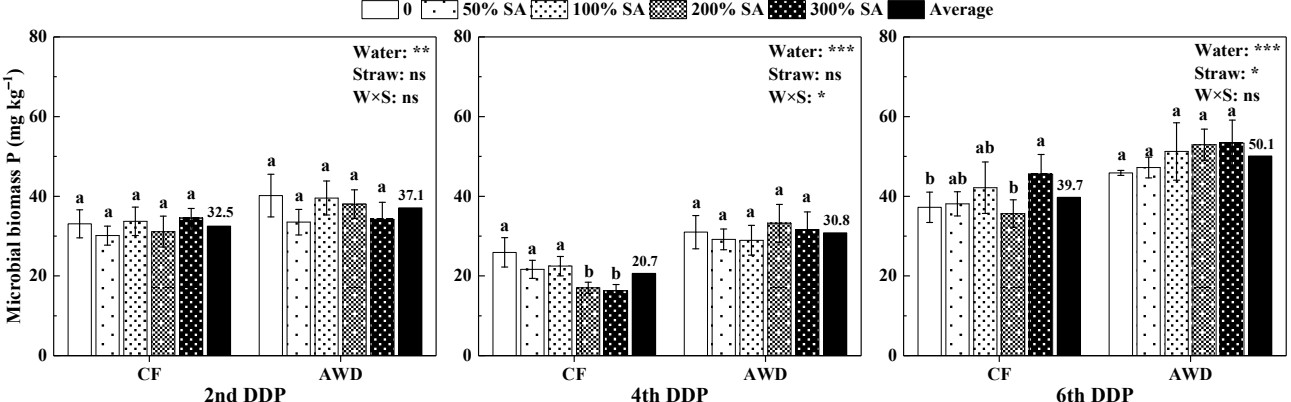

**Figure 4.** Soil microbial biomass P content under different water management regimes and straw addition rates. Different letters indicate significantly different at *p* < 0.05 under the same water management regime. Data in the figures indicate means of three replicates ± standard deviation. * means *p* < 0.05; ** means *p* < 0.01; *** means *p* < 0.001; ns means non-significant. CF, continuous flooding irrigation; AWD, alternate wetting and drying irrigation; SA, straw addition rate. DDP, day of the drying period.

### 3.3. Soil Redox Conditions

Soil total reductant content was significantly affected by water management (Figure 5). Compared with CF treatments, the average total reductant content of AWD treatments decreased by 9.47% and 18.5% on 4th DDP and 6th DDP, respectively. Under CF, soil total reductant content increased with straw addition rates, whereas straw addition had no significant effect on total reductant content under AWD. Water management and straw addition significantly affected soil ferrous iron content (Figure 6). Compared with CF treatments, the average ferrous iron content of AWD treatments decreased by 18.5% and 62.0% on 4th DDP and 6th DDP, respectively. During three sampling periods, ferrous iron content tended to increase with straw addition rates under two water management regimes.

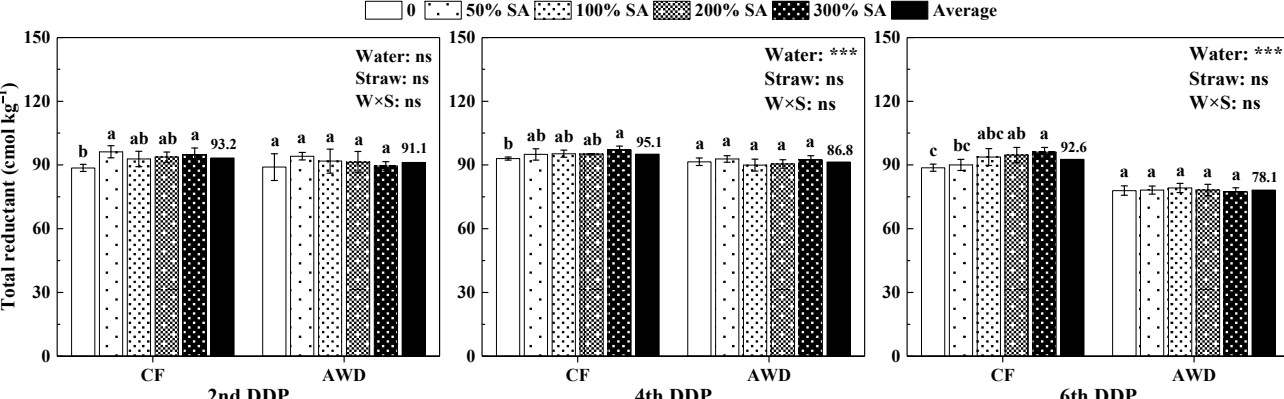

**Figure 5.** Soil total reductant content under different water management regimes and straw addition rates. Different letters indicate significantly different at *p* < 0.05 under the same water management regime. Data in the figures indicate means of three replicates ± standard deviation. *** means *p* < 0.001; ns means non-significant. CF, continuous flooding irrigation; AWD, alternate wetting and drying irrigation; SA, straw addition rate. DDP, day of the drying period.

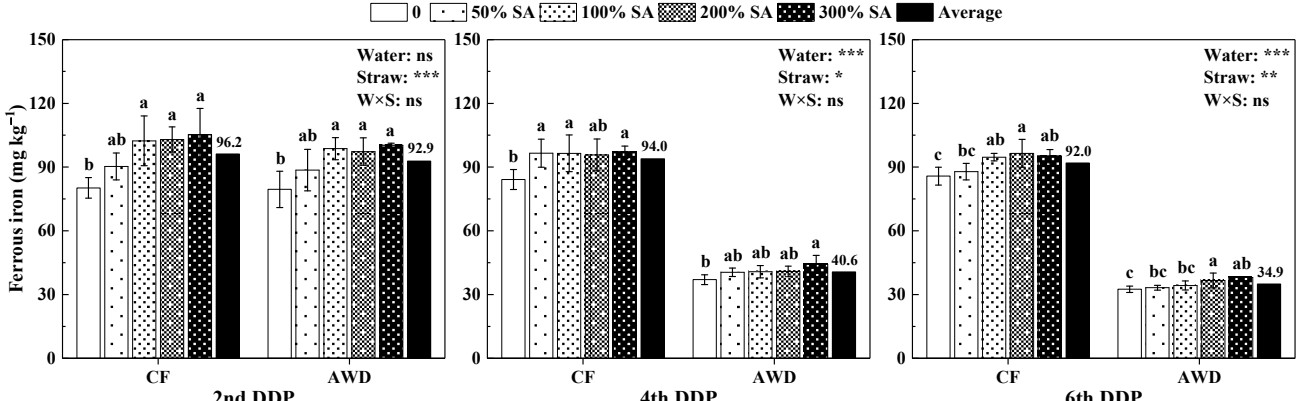

**Figure 6.** Soil ferrous iron content under different water management regimes and straw addition rates. Different letters indicate significantly different at *p* < 0.05 under the same water management regime. Data in the figures indicate means of three replicates ± standard deviation. * means *p* < 0.05; ** means *p* < 0.01; *** means *p* < 0.001; ns means non-significant. CF, continuous flooding irrigation; AWD, alternate wetting and drying irrigation; SA, straw addition rate. DDP, day of the drying period.

### 3.4. Relationships between Soil Water Content and Key Soil Properties under AWD

Soil water content was significantly correlated with available P content, microbial biomass P content, total reductant content, and ferrous iron content under AWD treatments (Figure 7). Soil available P content, total reductant content, and ferrous iron content increased with soil water content while the microbial biomass P content decreased. However, there was no significant correlation between soil water content and pH.

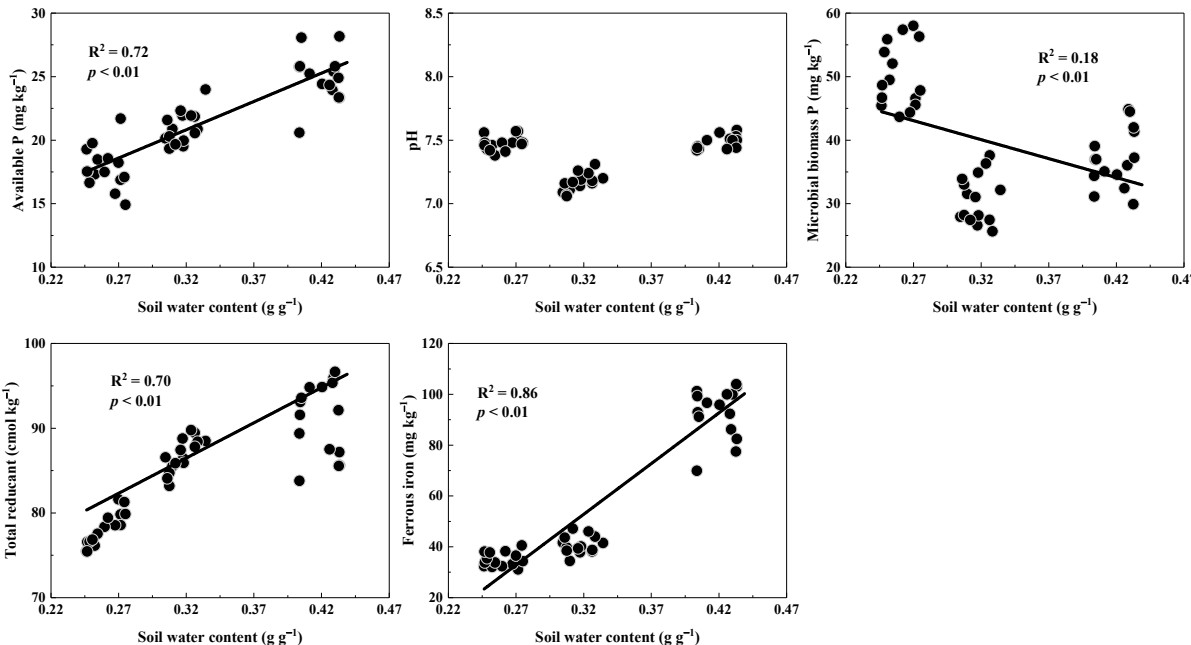

**Figure 7.** Relationships between soil water content and available P, pH, microbial biomass P, total reductant, and ferrous iron under alternate wetting and drying irrigation.

### 3.5. Correlations among Measured Soil Properties

Under CF, the available P content of fresh soil was not affected by other selected soil properties (pH, microbial biomass P, ferrous iron, and total reductant) (Table 1). However, soil available P content was positively correlated with total reductant content and ferrous iron content while negatively correlated with microbial biomass P content under AWD. Soil microbial biomass P content was positively correlated with soil pH while negatively correlated with total reductant content under CF and AWD. This indicated that strong reducibility decreased the activity of microorganisms. Soil ferrous iron content showed significant positive correlations with soil total reductant content and pH.

**Table 1.** Correlations between selected soil properties under continuous flooding irrigation (CF) and alternate wetting and drying irrigation (AWD).

| | | | | CF | | |
| | Indexes | AP | MBP | TR | FeII | pH |
|---|---|---|---|---|---|---|
| AWD | AP | | −0.019 | −0.60 | 0.061 | 0.133 |
| | MBP | −0.436 ** | | −0.343 * | −0.063 | 0.563 ** |
| | TR | 0.842 ** | −0.578 ** | | 0.279 | −0.015 |
| | FeII | 0.929 ** | −0.222 | 0.712 ** | | −0.120 |
| | pH | 0.184 | 0.598 ** | −0.098 | 0.382 ** | |

Note: AP, available P; MBP, microbial biomass P; TR, total reductant; FeII, ferrous iron. * means $p < 0.05$; ** means $p < 0.01$.

## 4. Discussion

The determined soil indexes under CF changed little except microbial biomass P content with three sampling periods, while the indexes under AWD changed obviously with soil water content (Figure 7). The change in soil water content leads to the change in redox potential [34]. Total reductant content, rather than Eh, which is difficult to accurately measure by a potentiometer due to different dry degrees of soil samples, can stably characterize the soil redox status. Soil flooding caused soil under anaerobic conditions, which increased total reductant content [13] and ferrous iron [8]. On the contrary, the decrease in soil water content made paddy soil gradually change from an anaerobic state to an aerobic state during the drying period of AWD. As a result, total reductant content and ferrous iron

content decreased with the decline of soil water content (Figure 7). Ferrous iron belongs to one kind of active, reducing substance, and our results showed that ferrous iron was more sensitive to water management and straw addition than total reductant (Figures 5 and 6).

Compared to CF, AWD decreased the available P content of fresh paddy soil under different straw addition rates at three sampling periods (Figure 2). This was ascribed to the decrease in soil ferrous iron content and the increase in microbial biomass P content. The decline of ferrous iron content suggested that more ferric ion was generated, which increased the formation of insoluble ferric phosphate through a precipitation reaction. On the other hand, reducing soil reducibility could stimulate microbial activity and then increase the retention of P by microorganisms. Furthermore, correlation analysis also suggested that available P was positively related to ferrous iron and negatively related to microbial biomass P under AWD (Table 1). The change in soil pH after flooding and drying was controlled by various factors, such as the initial soil pH, organic matter content, cation exchange capacity and flooding or drying time. Through a developed predictive model, Ding et al. [35] revealed that the pH of neutral-to-alkaline soils first decreased and then increased to approximately 7.0 during the flooding period while it increased linearly with the decreasing soil water content during the drying phase after flooding. This is not exactly consistent with our results. The pH values of almost all treatments at the three periods were in a neutral range of soil (pH 6.5–7.5) [36]. The change in the absolute value of $H^+$ concentration in this pH range was minimal, and the change in soil pH did not significantly affect the available P content (Table 1). These results confirmed our first hypothesis that AWD reduced paddy soil available P content compared with CF regardless of straw addition or not. However, Bagheri et al. found that AWD increased paddy soil's available P content, contrary to our results [4]. This may be due to the different soil types used in the experiment. The soil used in our investigation was neutral (pH = 7.06) while being alkaline (pH = 8.50) in their experiment. The fraction of P is dominated by iron-bound P (Fe-P) and aluminum-bound P (Al-P) in acidic to neutral soils, while it is dominated by calcium-bound P (Ca-P) in alkaline and calcareous soils [37].

In the present study, no significant effect of straw addition rates on soil available P was observed within the growing season under both water management regimes (Figure 1). It might be due to the short decomposition time of straw, which could not effectively form new organic matter, thus affecting soil P availability. The decomposition productions of new organic matter are beneficial for increasing P phytoavailability by affecting soil P absorption and/or decomposing organic P. Previous studies showed that soil organic matter could decrease P sorption through complexing the metal ions such as iron [38], competing for P adsorption sites, and affecting the binding energy of adsorbed P [39]. However, Guppy et al. suggested that the increase in available phosphorus content in sorption–desorption experiments is due to the P release from organophosphorus rather than the competitive adsorption sites of organic matter and P [40]. So, when the straw is completely decomposed, the available P content of soil with high straw addition rates maybe improved significantly. Moreover, the changing degrees of soil redox indexes (total reductant and ferrous iron) induced by straw addition were too small compared with water management. However, total reductant content under CF and ferrous iron content under two water management regimes showed an increasing trend with straw addition rates (Figures 5 and 6). If the straw returning amount in this study increases, it is possible to significantly change the P availability by vastly changing the soil redox status (especially ferrous iron). A 60-day incubation experiment with straw addition more than twice the maximum amount of our study showed that fresh soil available P content of straw addition treatment was significantly higher than control [4]. However, our experiment simulated field straw returning, and the 300% straw addition rate was large enough for the straw addition amount. Therefore, our findings cannot support our second hypothesis, and straw addition had no significant effect on soil P availability under both water management regimes within several weeks.

Available P content in fresh soil increased after air-drying under both water management regimes, which was consistent with the results of the previous study [4]. After air drying, the available P content of fresh soil increased, but the increment was not consistent between the two water management regimes at different sampling periods (Figure 1). The microbial biomass P of air-dried soil was also measured in the present study. However, most values of the air-dried samples were too low to be determined. Hence, the death of soil microorganisms may be an actual reason for the increase in available P after soil air drying [33]. After the death of microorganisms, the mineralized P may be partially fixed by soil clay and metal oxides, limiting the increase in available P in air-dried soil. Soil P availability may be affected by the mineralization of organic P by phosphatase [41] when soil water content from the water-holding capacity to air drying. During this period, P availability may also be affected by the transformation of $Fe^{2+}$ to $Fe^{3+}$ and increasing the formation of insoluble phosphate. The change in soil adsorption characteristics for P also changed P availability. This implies that the determination of available P under different water management regimes using air-dried soil could not reflect their differences.

Our research indicated that AWD decreased paddy soil P availability. However, some studies suggested that AWD had higher P use efficiency and rice yield than CF. This difference may be attributed to the lower water content and reducibility of paddy soil under the drying period of AWD. Such a soil environment is beneficial for promoting rice root growth [42] and root activity [43], thus increasing rice's nutrient uptake and utilization efficiency. Furthermore, a high concentration of ferrous iron under CF has toxicity, inhibiting rice growth and reducing grain yield through antagonistic effects on essential nutrients [44,45].

The limitation of the present study is that the soil samples were collected at different drying degrees of three wetting–drying cycles of AWD treatments. The responses of soil indexes to soil water content change may not be entirely consistent in the different wetting–drying cycles. Therefore, for further research, it can be considered to take samples every day for measurement in the same round of drying period of AWD. Then, the continuous change in available P content with drying degree can be evaluated. The field soil environment is more complex than the incubation experiment, and rice roots' oxygen secretion and root exudates exist in paddy soil. The water management and straw incorporation in field trials may be more likely to affect soil P availability.

## 5. Conclusions

Water management significantly affects paddy soil P availability. Compared to CF, AWD reduces soil available P content irrespective of straw addition. In the drying period of AWD, soil reducibility is weakened with the decrease in soil water content, while microbial activity is enhanced. All of these increase the fixation of soil P in the form of microbial biomass P, and ferrous iron content decreases, resulting in the reduction in available P content. Wheat straw with various addition rates show no significant effect on soil P availability within several weeks.

**Author Contributions:** Conceptualization, F.L.; methodology, X.Z.; software, Y.Z.; investigation, F.K. and X.Z.; resources, F.L.; data curation, F.K.; writing—original draft preparation, F.K.; writing—review and editing, F.K., X.Z., Y.Z., H.Y. and F.L.; visualization, F.L. and H.Y.; supervision, Y.Z.; funding acquisition, H.Y. All authors have read and agreed to the published version of the manuscript.

**Funding:** This work was supported by National Natural Science Foundation of China (No. 31770483).

**Institutional Review Board Statement:** Not applicable.

**Informed Consent Statement:** Not applicable.

**Data Availability Statement:** The data presented in this study are available upon request from the corresponding author.

**Conflicts of Interest:** The authors declare no conflict of interest.

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
