# Peer review of "Alternate Wetting and Drying Irrigation Reduces P Availability in Paddy Soil Irrespective of Straw Incorporation"

_agronomy, doi:10.3390/agronomy12071718_

Round 1

Reviewer 1 Report

General comments

Phosphorus is a major nutrient for crop production and soil P availability is highly influenced by soil properties including soil moisture, pH and organic matter. This manuscript presents the results from an incubation study on the effect of flooding and alternate wetting and drying cycle and their interaction with various rates of straw incorporation on the influence. The study is scientifically sound and the methodology is robust. The data generated is publishable. The manuscript needs some revisions to improve the standard. Overall, the manuscript needs some improvement in language.

Specific comments

Line 141-143- Give reference to the P analysis method. An explanation of the anti-colorimetric method will improve the clarity. Do authors mean colorimetric?

Discussion

The discussion will benefit from incorporating the discussion on soil reactions in flooded soils versus the soil reactions during the drying cycle after flooding. What happens to soil pH upon flooding and what happens to soil pH during the dry phase after flooding. What is the relationship to inherent soil pH after flooding and what happens during drying and how does that impact soil P availability?

Also, a discussion on the P content of organic matter added (straw) and what difference will it make to soil available P even if 100% straw breakdown.

The discussion on sorption-desorption reaction to the organic matter will improve by referring to the following reference- Guppy et al 2004.

https://www.publish.csiro.au/sr/sr04049

Author Response

Dear Reviewer:

Thank you for your comments concerning our manuscript (agronomy-1792706) entitled “Alternate wetting and drying irrigation reduces P availability in paddy soil irrespective of straw incorporation”. Those comments are valuable and very helpful for revising and improving our manuscript. We have studied comments carefully and have made correction which we hope meet with approval. Revised portion are marked in red in the manuscript. The main corrections are as flowing:

Overall, the manuscript needs some improvement in language.

Response: We have changed it as suggested.

Line 141-143- Give reference to the P analysis method. An explanation of the anti-colorimetric method will improve the clarity. Do authors mean colorimetric?

Response: Yes. And we have added the reference and described it in more detail in line 151-155.

The discussion will benefit from incorporating the discussion on soil reactions in flooded soils versus the soil reactions during the drying cycle after flooding.

Response: We discussed these soil reactions in line 271-275, line 279-283.

What happens to soil pH upon flooding and what happens to soil pH during the dry phase after flooding. What is the relationship to inherent soil pH after flooding and what happens during drying and how does that impact soil P availability?

Response: Thank you for your suggestions. We have tried to discussed these in line 287-293. Because the change of soil pH is complex and controlled by various factors, such as the initial soil pH, organic matter content, cation exchange capacity and flooding or drying time. Our findings of soil pH didn’t exactly match the results of previous study. And according to the definition of pH, soil pH in a range of 7.15-7.61 in our study actually has a small variation range of soil hydrogen ion concentration. We believe that soil pH is not the main factor affecting the P availability in this experiment. 

Also, a discussion on the P content of organic matter added (straw) and what difference will it make to soil available P even if 100% straw breakdown.

Response: We thought soil available P with high straw addition rates will be significant increased if 100% straw breakdown. And we have added the related discussion in line 307-316.

The discussion on sorption-desorption reaction to the organic matter will improve by referring to the following reference- Guppy et al 2004.

Response: Thanks for providing a new reference for us to refer to. It was helpful to improve the quality of our manuscripts. We have improved our discussion in line 308-315.

Yours sincerely,

Fengmin Li

12 July. 2022

Reviewer 2 Report

The paper is interesting and the experimental design well structured. the introduction section should be rewritten because it appears to be a list of important aspects that should be better connected to each other. The text must be reviewed by a native English speaker. Here are some details:

- line 13: days instead of "d"

- line 24: remove a full stop at the end of the sentence

- line 73: here and elsewhere check 3+ as superscript

- line 130: are you sure that the centrifugation of the sample has not also removed part of the soluble P? Explain this methodological aspect better

- line 164: remove a parenthesis after DDP

Author Response

Dear Reviewer 2:

Thank you for your comments concerning our manuscript (agronomy-1792706) entitled “Alternate wetting and drying irrigation reduces P availability in paddy soil irrespective of straw incorporation”. Those comments are valuable and very helpful for revising and improving our manuscript. We have studied comments carefully and have made correction which we hope meet with approval. Revised portion are marked in red in the manuscript. The main corrections are as flowing:

The introduction section should be rewritten because it appears to be a list of important aspects that should be better connected to each other.

Response: We have rewritten the introduction and tried to make the content of the sections more connected.

The text must be reviewed by a native English speaker. 

Response: We have changed it as suggested.

line 13: days instead of "d" 

Response: Thank you for pointing this out. We have changed "d" to "days".

line 24: remove a full stop at the end of the sentence

Response: Thank you for pointing this out. We have changed it as suggested.

line 73: here and elsewhere check 3+ as superscript 

Response: Thank you for pointing this out. We have changed it as suggested.

line 130: are you sure that the centrifugation of the sample has not also removed part of the soluble P? Explain this methodological aspect better 

Response: It does lose part of the soluble P. But the content of water-soluble P in paddy soil is usually very low. According to a study, the concentration of total P in surface water in paddy is 0.76 mg L-1 (Lu et al., 2018). We believe that low centrifugation rate using in the method and the loss of the water-soluble P content have negligible effects on our experimental results. And We have supplemented the reason for using centrifugation in line 133-137.

http://www.ere.ac.cn/EN/10.11934/j.issn.1673-4831.2018.04.008   --- Lu et al., 2018

line 164: remove a parenthesis after DDP

Response: We think the parenthesis is needed in this sentence. Please check it again.

Yours sincerely,

Fengmin Li

12 July. 2022

Reviewer 3 Report

In general the article is fine. However, there are a number of issues that I think the authors must improve so that the article can be published.

Here are some of the improvements you should consider

In reference to the Straw management there is a confusion or a misunderstanding from my point of view:

In line 105 à “were filled with equivalent to 400 g dry soil, which mixed uniformly 105 with the corresponding wheat straws” but in line 127 à “After removing the straw and mixing soil samples uniformly, the soil water content 127 of AWD treatments was measured directly by the oven drying method”. How can authors remove the Straw if it is cut to 1-2 cm and is mixed and incorporated with the soil?

I believe that authors should improve the straw addition and how they separated soil and straw.

In my opinion text between lines 128 – 132 should be improved: “Figure 1 shows the soil water content of AWD treatments at three sampling periods. The soil water content of AWD treatments was consistent in each sampling period. Soils of CF treatments were centrifuged for 10 mins (3600 r/min) to remove surfaced water to make their soil water content as consistent as possible. The average soil water content of CF treatments was 0.466 g g-1.”

How many measurements were carried out by each treatment in each DDP? The CF was centrifuged, but how was determined in the case of AWD treatments?

The average of CF is clear but what does it happen in the case of AWD? Is “0.466” an average of different periods or a data from a specific DDP?

How many samples were carried out in the experiment? Authors explain in the article that “The experimental design was a factorial combination of water management and 103 straw addition”, I suppose that there is 2 water management treatment and 5 straw combinations, but there is not specified in the text. How many repetitions have made? Reformulate

Did authors test the normality of data?

Lines 162-163: “Both available P content of fresh and air-dried soil were significantly affected by water management (except fresh soil available P content on 2nd day of the drying period (DDP))”. à Why did authors detect a difference in the 2nd day? Can explain it?

In the figures you use a CK data. what does it mean? In the text and captions there is not explained.

Figures: In my opinion, letters to identify the significant differences should be indicated in all the graphics, there are some of them with letters and other without them.

Author Response

Dear Reviewer:

Thank you for your comments concerning our manuscript (agronomy-1792706) entitled “Alternate wetting and drying irrigation reduces P availability in paddy soil irrespective of straw incorporation”. Those comments are valuable and very helpful for revising and improving our manuscript. We have studied comments carefully and have made correction which we hope meet with approval. Revised portion are marked in red in the manuscript. The main corrections are as flowing:

In line 105 à were filled with equivalent to 400 g dry soil, which mixed uniformly 105 with the corresponding wheat straws but in line 127 à After removing the straw and mixing soil samples uniformly, the soil water content 127 of AWD treatments was measured directly by the oven drying method . How can authors remove the Straw if it is cut to 1-2 cm and is mixed and incorporated with the soil? I believe that authors should improve the straw addition and how they separated soil and straw.

Response: Even after 40 days of incubation, the breakdown of wheat straw is not high, and the 1-2 cm straw segments still remain intact. We can use tweezers to pick them out of the soil. We have added “remove the straw from soil by using tweezers” in Line 129.

In my opinion text between lines 128 132 should be improved: Figure 1 shows the soil water content of AWD treatments at three sampling periods. The soil water content of AWD treatments was consistent in each sampling period. Soils of CF treatments were centrifuged for 10 mins (3600 r/min) to remove surfaced water to make their soil water content as consistent as possible. The average soil water content of CF treatments was 0.466 g g-1. How many measurements were carried out by each treatment in each DDP? The CF was centrifuged, but how was determined in the case of AWD treatments? The average of CF is clear but what does it happen in the case of AWD? Is 0.466 an average of different periods or a data from a specific DDP?

Response: Thank you for pointing this out. Three replicates were carried out by each treatment in each DDP. The soil water content of AWD treatments was measured directly by the oven drying method (Line 130-131). 0.466 is an average of different periods of CF. The difference of soil water content after centrifugation of CF as well as three replicates of each AWD treatment in a specific DDP is very small. And we recorded the soil water content of each sample to calculate the contents of chemical indexes. We have described them in more detail in Line 136-137, 140, 145-146.

How many samples were carried out in the experiment? Authors explain in the article that The experimental design was a factorial combination of water management and 103 straw addition , I suppose that there is 2 water management treatment and 5 straw combinations, but there is not specified in the text. How many repetitions have made? Reformulate

Response: Thank you for pointing this out. Three repetitions have made. We have changed these in Line 105, 112-113, 126.

Did authors test the normality of data?

Response: Yes, we did. We have described it in more detail in Line 167-169.

Both available P content of fresh and air-dried soil were significantly affected by water management (except fresh soil available P content on 2nd day of the drying period (DDP)) . à Why did authors detect a difference in the 2nd day? Can explain it?

Response: We may not have accurately understood the reviewer's question. Air-dried soil available P was significantly affected by water management on 2nd DDP. It might be due to the death of microorganisms which increase the available P after soil air drying. We could see that the MBP of AWD treatments are higher than CF on 2nd DDP (Figure 4). Moreover, although available P-fresh has no effects under water management, during air drying stage, phosphatase, P sorption capacity of soil, the change of Fe2+ and so on may affect P availability of air-dried soil (Line 339-343).

In the figures you use a CK data. what does it mean? In the text and captions there is not explained.

Response: Thank you for pointing this out. We have changed “CK” to “0” in the figures.

Figures: In my opinion, letters to identify the significant differences should be indicated in all the graphics, there are some of them with letters and other without them.

Response: We have changed it as suggested.

Yours sincerely,

Fengmin Li

12 July. 2022

Round 2

Reviewer 1 Report

The authors have clarified the reviewers' queries and improved the manuscript.

Reviewer 3 Report

Thanks to the authors for the improvement! The authors explained the question about the P in lines 339 - 343, as they mentioned. I exposed that question because when I read the structure of the paper, in my mind is expected that if the first outcome is P available, in the discussion, it appears as the first, but it is indeed discussed in that lines. Sorry for the confusion.